# High Drug Capacity Doxorubicin-Loaded Iron Oxide Nanocomposites for Cancer Therapy

Ekaterina Kovrigina , Alexey Chubarov * and Elena Dmitrienko *

Institute of Chemical Biology and Fundamental Medicine SB RAS, 630090 Novosibirsk, Russia;
e.bobrikova96@gmail.com

* Correspondence: chubarov@niboch.nsc.ru or chubarovalesha@mail.ru (A.C.); elenad@niboch.nsc.ru (E.D.);
Tel.: +7-913-763-1420 (A.C.); +7-913-904-17-42 (E.D.)

**Abstract:** Magnetic nanoparticles (MNPs) have great potential in the drug delivery area. Iron oxide ($Fe_3O_4$) MNPs have demonstrated a promising effect due to their ferrimagnetic properties, large surface area, stability, low cost, easy synthesis, and functionalization. Some coating procedures are required to improve stability, biocompatibility, and decrease toxicity for medical applications. Herein, the co-precipitation synthesis of iron oxide MNPs coated with four types of primary surfactants, polyethylene glycol 2000 (PEG 2000), oleic acid (OA), Tween 20 (Tw20), and Tween 80 (Tw80), were investigated. Dynamic light scattering (DLS), ζ-potential, and transmission electron microscopy (TEM) techniques were used for morphology, size, charge, and stability analysis. Methylene blue reactive oxygen species (ROS) detection assay and the toxicity experiment on the lung adenocarcinoma A549 cell line were conducted. Two loading conditions for anticancer drug doxorubicin (DOX) on MNPs were proposed. The first one provides high loading efficiency (~90%) with up to 870 µg/mg (DOX/MNPs) drug capacity. The second is perspective for extremely high capacity 1757 µg/mg with drug wasting (DOX loading efficiency ~24%). For the most perspective MNP_OA and MNP_OA_DOX in cell media, pH 7.4, 5, and 3, the stability experiments are also presented. MNP_OA_DOX shows DOX pH-dependent release in the acidic pH and effective inhibition of A549 cancer cell growth. The IC50 values were calculated as $1.13 \pm 0.02$ mM in terms of doxorubicin and $0.4 \pm 0.03$ µg/mL in terms of the amount of the nanoparticles. Considering this, the MNP_OA_DOX nano theranostics agent is a highly potential candidate for cancer treatment.

**Keywords:** iron oxide nanoparticles; functionalization; coating; polyethylene glycol; oleic acid; tween; drug delivery; doxorubicin; toxicity

## 1. Introduction

Iron oxide magnetic nanoparticles (MNPs) have become a valuable tool for structural biology, biochemistry, diagnostics, and medical applications [1–7]. MNPs are cost-effective and excellent MRI contrast agents, hyperthermia therapy agents, and external magnetic field-influenced transport systems [2,7–15]. Nanomaterials have been used widely for drug delivery to improve the therapeutic index and avoid side effects. Combining approaches of MRI diagnostics, hyperthermia therapy, and drug loading have a great potential for directed cancer treatment [4–6,16]. Magnetite nanoparticles ($Fe_3O_4$) have the most perspectives due to their ferrimagnetic properties. However, such MNPs are not stable upon oxidation to $Fe_2O_3$ and possess high surface energy, leading to aggregation. One more problem is the potential cytotoxicity due to the reactive oxygen species (ROS) formation [17–19]. Thus, surface coating or functionalization is required for magnetic core protection, colloidal stability increase, and the toxicity effect decrease.

A modern approach uses tannic acid, classical carboxylic acid (polyacrylic, lauric, myristic, or oleic), hyaluronic acid, Tween (polysorbate), polyethylene glycol (PEG), polyethyleneimine, dextran, chitosan, organic polymers, and proteins for colloidal stabilization and optimal magnetic nucleus size formation [4,7,20–34]. For example, PEG-coating

enhances the colloidal dispersion and stability of MNPs [4,20,35]. PEGylation decreases the nonspecific protein adsorption and phagocyte uptake, prolongs blood circulation time, and improves biocompatibility [4]. Oleic acid (OA) and other fatty acids are commonly used surfactants to stabilize the MNPs [25,26]. They are strongly bound by carboxylic acid to the iron oxide surface. It is necessary for making monodisperse and highly uniform nanoparticles. Tween (Tw) is a non-ionic surfactant commonly used in drug delivery areas for dispersing hydrophobic drugs. It consists of a hydrophilic head group and a hydrophobic alkyl chain. Based on alkyl tail length, several Tween surfactants are possessed to choose the hydrophile–hydrophobic balance [20,36].

"Smart" nanoparticles with pH-stimuli-responsive drug release are promising candidates for cancer treatment. A combination of magnetic properties, hyperthermia, MRI, and other features of MNPs with drug-loading shows great potential as theranostic nanocarriers for tumor-targeted delivery and diagnostics. Doxorubicin (DOX) is one of the most widely used chemotherapeutic drugs for treating various types of cancer [37,38]. It is an FDA-approved anthracycline chemotherapeutic that has been used for more than 40 years. Unfortunately, DOX is often associated with multidrug resistance and many side effects [37,38]. One of the prospective possibilities to avoid drug resistance is a synthesis of DOX-loaded nanoparticles [39–46]. Several constructions were studied for DOX loading. Among them are gold [45], carbonate [40], iron oxide nanoparticles [21,39,41,42,47–59], metal-organic frameworks [60], and much more laborious and complex constructions [61–64]. However, MNPs in the published works demonstrate only a low noncovalent drug loading capacity (µg/mg, DOX/MNP): 11 [55,56], 22 [57], 50 [42,51,58], 91 [54], 110 [41], 120 [65,66], 131 [50], 300 [53]. Moreover, for further extended biological research on cell and mouse models, the excellent capacity of DOX is required. Along with the high DOX-loaded MNPs stability at physiological pH ~ 7.4, the pH-dependent drug release in the tumor microenvironment and endosomal cell compartment with acidic pH ~ 5–5.5 is essential.

Herein, we present the one-pot co-precipitation of iron oxide magnetic nanoparticles coated with PEG 2000, OA, and Tw surfactants (Figure 1). Tw 20 and Tw 80 compounds with a high difference between hydrophobic chain lengths were chosen. The prepared MNPs were investigated by transmission electron microscopy (TEM) and dynamic light scattering (DLS) methods. The nanoparticles' size and $\zeta$-potential were measured. The particles' morphological properties and agglomerates' formation were also investigated by TEM. MNPs' ROS formation was calculated by a methylene blue (MB) dye decolorization assay. We demonstrate that the resulting MNPs are biocompatible, which is demonstrated by low ROS formation and toxicity experiments using a lung adenocarcinoma A549 cell line. The perspective MNPs were tested on an anticancer drug DOX loading efficiency. Two possible drug loading conditions were proposed at a basic pH. The first provides excellent capacity, up to 1757 µg DOX/1 mg for MNP_OA, with a 24% DOX loading efficiency. Up to 870 µg DOX/1 mg for MNP_OA with magnificent (~90%) DOX loading efficiency was found for the second approach. The stability experiments of MNP_OA and perspective DOX-loaded OA-coated MNP (MNP_OA_DOX) samples in cell media, pH 7.4 in FBS, and a low pH of 3 and 5 are also presented. A negligible amount of doxorubicin was released at a physiological pH (7.4), confirming the pH-dependent manner of drug release. At an acidic pH (5, similar to that found in the tumor environment or the endosomal compartment), MNP_OA_DOX shows a ~70% DOX release for 3 h. MNP_OA_DOX retains high stability in cell media and good inhibition of cancer A549 cell proliferation. The presented synthesis and high-capacity drug-loading method increase the therapeutic response of DOX-loaded MNPs in cancer cells. The use of the carrier features and diagnostic possibilities with improved DOX loading may highly increase the contribution of MNPs to the drug-resistance cancer treatment.

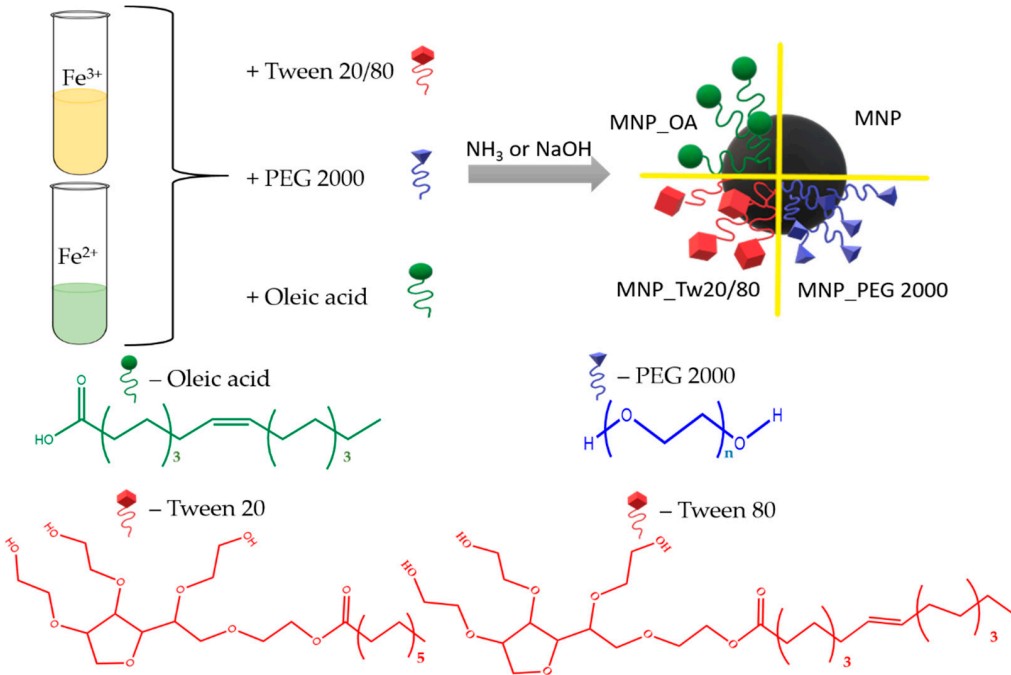

**Figure 1.** Representation of Tw, OA, or PEG-coated magnetic nanoparticle synthesis.

## 2. Results and Discussion

### 2.1. Synthesis and Characterization of MNP

Magnetic iron oxide nanoparticles were synthesized by the classical co-precipitation method. The co-precipitation technique is among the simplest and most cost-effective of MNPs' synthesis procedures. Despite the advantages such as high yield, high product purity, and the lack of necessity to use organic solvents, the easily reproducible MNPs obtained by this method are often not stable. Moreover, some surfactants and clear reaction parameters are required for the MNPs' stability and reproducibility. The procedure consists of $Fe^{2+}/Fe^{3+}$ salts and surfactant co-precipitation under base conditions. The synthesis scheme is shown in Figure 1. Several types of widely used surfactants were chosen to protect and impart surface properties. Among them are the amphiphilic polyethylene glycol 2000 (PEG 2000) [16,25,53], hydrophobic anionic oleic acid (OA) [67], and Tween (Tw) family [20,68].

Such surfactants may lead to surface stabilization, colloidal stability, and reduce the iron (II) oxidation [4,16,20,25,26,35,36,69]. Moreover, the selected surfactants are approved by the FDA for medical application, which confirms their potential [36]. For comparison, a non-coated MNP by the procedure previously published by our group was synthesized [70]. The resulted nanoparticles were characterized by TEM and DLS methods (Table 1, Figure 2). The surface charge and size of nanoparticles are the major factors that influence cell internalization, tissue delivery, and renal clearance. Small nanoparticles smaller than 5–10 nm have demonstrated high organism clearance by the renal system, which is not optimal for cancer treatment. A too high nanoparticle size (>200 nm) has a problem passing through cellular membranes and organism transport in comparison with a smaller size [35]. The size data obtained by DLS and TEM methods for MNP, MNP_OA, and MNP_PEG 2000 are in a good correlation (Table 1). The DLS figures (number and intensity modes) are presented in the Supplementary Materials (Figures S1–S9).

**Table 1.** Hydrodynamic diameter, polydispersity index (PDI), and $\zeta$-potential of synthesized MNPs in water (pH 5.5).

| Sample | Diameter by TEM, nm | Hydrodynamic Diameter by DLS, nm | PDI | $\zeta$-Potential, mV |
|---|---|---|---|---|
| MNP | $10 \pm 2.5$ | $15.6 \pm 2.2$ | $0.296 \pm 0.003$ | $36.0 \pm 0.9$ |
| MNP_Tw20 | - | $258 \pm 4$ | $0.44 \pm 0.01$ | $37.0 \pm 0.8$ |
| MNP_Tw80 | - | $233 \pm 2$ | $0.256 \pm 0.007$ | $36.0 \pm 1.5$ |
| MNP_PEG 2000 | $200 \pm 50$ | $196 \pm 15$ | $0.50 \pm 0.03$ | $27.0 \pm 3.0$ |
| MNP_OA | $100 \pm 40$ | $112 \pm 19$ | $0.172 \pm 0.009$ | $-43.0 \pm 0.8$ |

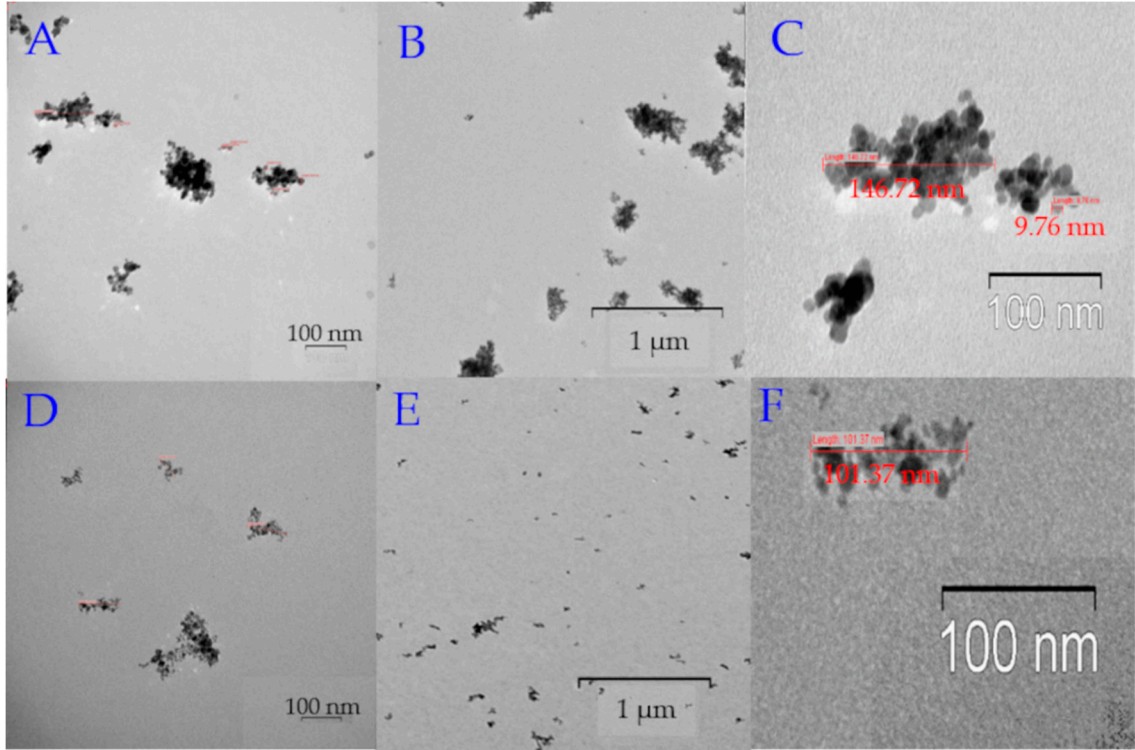

**Figure 2.** TEM images of MNPs. (**A**) High magnification TEM image MNP_PEG 2000; the bar indicates 100 nm. (**B**) Low magnification TEM image MNP_PEG 2000; the bar indicates 1 μm. (**C**) TEM image reveals the MNP_PEG 2000 size (~150–200 nm). (**D**) High magnification TEM image MNP_OA; the bar indicates 100 nm. (**E**) Low magnification TEM image MNP_OA; the bar indicates 1 μm. (**F**) TEM image reveals the MNP_OA size (~100–110 nm).

MNP_OA forms ~10 nm sizes grouped into larger aggregates (~100 nm) (Figure 2D–F). Groups of the images in Figure 2A–F show the same pictures with different magnification. It should be noted that regions with low electron density are visualized between nanoparticles, which indicates no accumulations of nanoparticles, but the grouping of particles are in the sample preparation for TEM. The size ($112 \pm 19$ nm, PDI = $0.172 \pm 0.009$) and $\zeta$-potential ($-43 \pm 2$ mV) of MNP_OA were determined by DLS. Unlike other synthesized MPNs, MNP_OA has a negative $\zeta$-potential due to the negative charge of the oleic acid carboxyl group. MNP_PEG 2000 nanoparticles are much larger than MNP_OA. The size is equal to $196 \pm 15$ nm, with a PDI of $0.5 \pm 0.03$ and a $\zeta$-potential of $27 \pm 3$ mV. It was found by TEM (Figure 2A–C) that MNPs are presented as individual clusters with a maximum size of up to 250 nm. MNP_OA and MNP_PEG 2000 show good colloidal stability for no less than 2–3 weeks (see Section 2.4). MNP_Tw samples form high clusters with a size of about 250 nm and a positively charged surface of ~30 mV (Table 1). TEM images could not be obtained due to difficulties in the sample preparation. MNP_Tw quickly precipitates to the bottom of the container. It is necessary to act on the particles with an ultrasound for

15 min to restore their colloidal behavior. MNP_Tw is not stable during storage, forms large aggregates (more than 3 µm), and settles to the bottom of the falcon. That is why MNP_Tw was excluded from further investigations.

### 2.2. Reactive Oxygen Species (ROS) Formation Study

Iron ions may effectively produce ROS in the human organism [17]. In this way, some biocompatibility studies are required for iron oxide MNPs. Under organism conditions, ferrous ion release is possible, which leads to the disruption in the activity of ion channels, cytoskeleton function, ROS formation, etc. One of the easy assays to calculate ROS generation is the methylene blue (MB) decolorization test using hydrogen peroxide [71–73].

The ROS detection assay results are presented in Figure 3 and Figure S11. In Figure 3, the data after 10 min is shown. The kinetics graphs can be found in Figure S11. The percentage of dye binding on the MNPs' surface was subtracted from the percentage of degradation of MB in the reaction with MNP and $H_2O_2$. No significant MB affinity to MNPs was detected. The results are in good correlation with the previously obtained data [70].

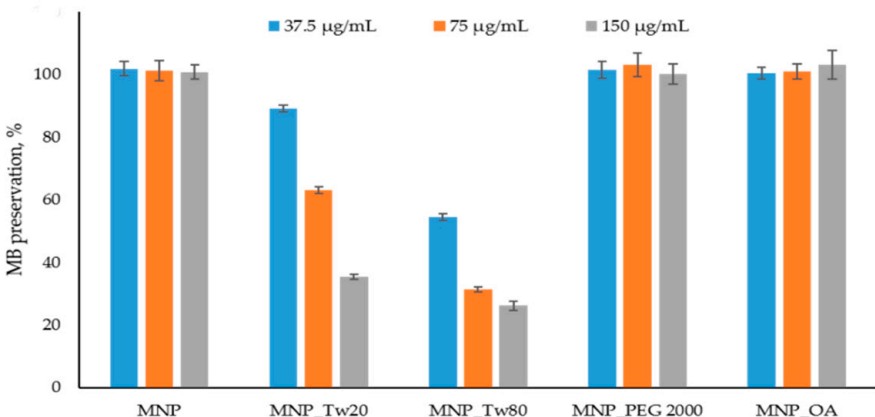

**Figure 3.** Methylene blue (MB) decolorization assay. The MB degradation % was calculated spectrophometrically (see Section 3.6). The reactions were carried out using 37.5 µg/mL, 75 µg/mL, and 150 µg/mL MNPs and control solution without nanoparticles. No significant peroxide degradation was obtained in the presence of MNPs.

MNP, MNP_OA, and MNP_PEG 2000 show no MB degradation for at least 1.5 h at a 75 µg/mL concentration (Figure S11). The result for non-coated MNP is in good correlation with previously published data [70]. Unfortunately, MNP-Tw nanoparticles are not perspective for further investigation because of the ROS formation and possible high toxicity (Figure 3). Using a very high concentration of MNPs (150 µg/mL), the sensible decrease in MB to 60% is observed only for MNP_PEG 2000 after 1.5 h (data not shown). Therefore, for further experiments, MNP_OA and MNP_PEG 2000 samples were used.

### 2.3. Anticancer Drug Doxorubicin Loading

The drug loading efficiency onto the MNPs carrier is also an important parameter. Drugs can be loaded onto nanoparticles' surfaces using many procedures [39–46]. To evaluate the drug loading potential of MNPs, a chemotherapeutic drug, DOX, with broad-spectrum antitumor activity was selected. Moreover, DOX has a fluorescence property, which confers a convenient drug loading traceability. For this purpose, we have chosen the sodium borate buffer SBB (pH 8.5), since the DOX sorption increases in alkaline conditions [40,43,45]. The loading of doxorubicin (DOX, 1 mg/mL) on the MNPs (1 mg) was carried out in 1 mL of sodium borate buffer (SBB, 10 mM, pH 8.5) under 25 °C (Figure 4).

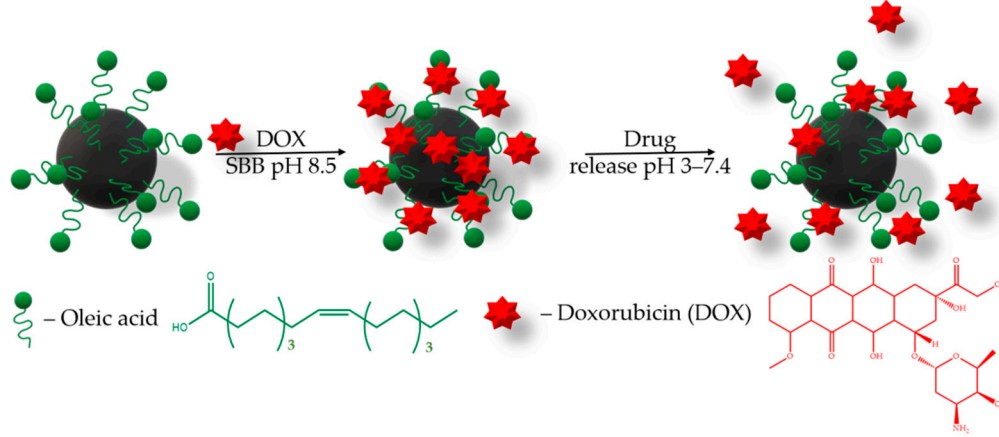

**Figure 4.** Schematic description of the DOX loading onto MNPs and release.

The adsorption of DOX on MNPs was studied by UV-vis. spectroscopy (see Section 3.8). The amount of the drug bound to the MNPs was calculated as the difference between added DOX and DOX remaining in the solution after the incubation with nanoparticles. The capacity data are presented in Table 2. No question is raised about MNPs having excellent DOX loading capacities. However, non-coated MNPs have negligible DOX loading capacity (Table 2). The presence of a shell of surfactants, such as PEG 2000 and OA, increases the sorption capacity. MNP_OA and MNP_PEG 2000 have good capacity values according to the literature data [39–46]. The increase in capacity is most likely associated with the presence of a surfactant shell, which forms a porous coating that provides more vacancies for a drug interaction with coated MNPs.

**Table 2.** The comparison of the DOX loading MNP capacity.

| Sample | DOX/MNP μg/mg | DOX Loading Efficiency [b] |
|---|---|---|
| MNP | n.d. [a] | n.d. |
| MNP_PEG 2000 | $590 \pm 10$ | 59% |
| MNP_OA | $868 \pm 37$ | 87% |

[a] Not determined, negligible amount of DOX. [b] DOX loading efficiency = DOX in the solution after loading/initial amount of DOX; 1 mL of DOX solution (1 mg/mL) per 1 mg of MNPs was used.

The physical characteristics of the drug and MNPs' coating are also a key factors for drug loading. DOX is a principally hydrophobic drug with a chemical group that can form electrostatic interaction and hydrogen bonding. PEG 2000 is an amphiphilic compound that can, on one side, form some hydrogen bonding and noncovalent van der Waals interactions. Oleic acid is a highly hydrophobic compound that can highly bind the anthracycline part of the DOX. The increase in the capacity of MNP_OA relative to MNP_PEG 2000 may be associated with the changes in the nanoparticle charge from the positive (MNP_PEG 2000 and $\zeta$-potential = $27 \pm 3$ mV) to negative (MNP_OA and $\zeta$-potential = $-43 \pm 2$ mV) [42]. DOX has a positively charged amino group (pKa = 8.3) under loading conditions (pH 8.5). For the DOX-loaded MNP_OA_DOX, the hydrodynamic diameter increases to 201 nm and the $\zeta$-potential becomes lower, at $-8.3$ mV (Table 3).

In the above presented MNPs' capacity experiment, the loading efficiency rises to 90% (Table 2). However, under such limiting conditions, the highest capacity of MNPs may not be obtained. The experiment with various MNPs' amounts and the same quantity of DOX is presented in Figure 5 and Table 4. The results suggest the extraordinary capability of MNP_OA to load high contents of DOX at a base pH. However, the drug loading efficiency is much lower than in the previous experiment (cf. Tables 2 and 4). Further experiments are required to develop optimal loading conditions.

**Table 3.** Hydrodynamic diameter, polydispersity index (PDI), and ζ-potential of synthesized MNP_OA_DOX in water.

| Sample | Hydrodynamic Diameter by DLS, nm | PDI | ζ-Potential, mV |
|---|---|---|---|
| MNP_OA | 112 ± 19 | 0.172 ± 0.009 | −43.0 ± 0.8 |
| MNP_OA_DOX | 201 ± 51 | 0.216 ± 0.030 | −8.3 ± 0.2 |

**Table 4.** DOX loading MNP_OA capacity in sodium borate buffer (SBB, 10 mM, pH 8.5).

| MNP_OA, mg | DOX/MNP, μg/mg | DOX Loading Efficiency * |
|---|---|---|
| 0.069 | 1757 ± 108 | 24.3% |
| 0.138 | 944 ± 30 | 26.1% |
| 0.274 | 610 ± 16 | 33.7% |
| 0.690 | 333 ± 19 | 46.0% |

* 0.5 mL of DOX solution (1 mg/mL) per MNPs was used.

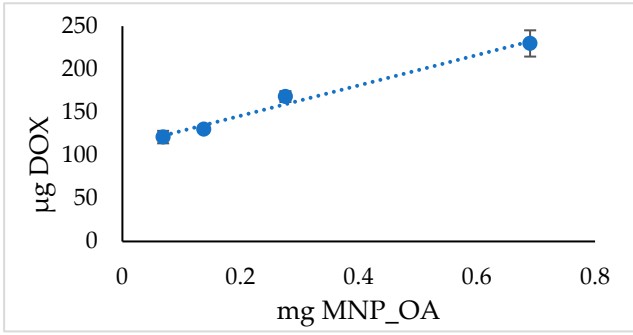

**Figure 5.** DOX loading MNP_OA capacity.

*2.4. MNP_OA and MNP_OA_DOX Stability in Aqueous Solution*

The nanoparticle stability in an aqueous solution is an important factor for biomedical applications. Oleic acid (OA) is often used because it forms a protective monolayer for making monodisperse MNPs [74]. The surface of such MNPs is usually very hydrophobic. However, according to the procedure [75], it is possible to obtain water-soluble and stable MNPs. In our experience, deionized water (Milli-Q) instead of distilled water is necessary for the stage of nanoparticle synthesis. However, the size of nanoparticles is a bit varying (±15–20 nm) from one synthesis to another due to the co-precipitation method, which is highly dependent on the reaction parameters (pH, temperature, mixing speed, etc.). After the synthesis, MNP_OA retains colloidal stability during 2–3 weeks at pH > 7 under 4 °C without significant changes in the size evaluated by DLS (data not shown). Figure 6 shows the stability investigations of MNP_OA and MNP_OA_DOX under several conditions: (1) Milli-Q water; (2) cell medium DMEM (pH ~ 7); (3) acetic buffers (pH 3 and 5).

Cell medium with neutral pH recreates physiological conditions in which MNPs have to be stable, and no drug release has occurred. An acetic buffer of pH 5 mimics the tumor microenvironment or cell endosomal compartment. The pH in the tumor extracellular matrix is often more acidic than the pH in normal tissues and blood. Moreover, the endosomes, in which the nanoparticles will be in the cell, usually have acidic pH ~ 5.5–5.0. An acetic buffer of pH 3 shows the extreme, which is impossible for the cell option. Based on these results in Figure 6 and Tables S1–S4 (DLS size and PDI data), we can conclude that MNP_OA and MNP_OA_DOX are stable not less than for 48 h in the most often presented condition. In the acetic buffer (pH 3), MNP_OA is not stable. The aggregation of MNPs occurred after several minutes. Tw20 solution was added to improve its colloidal stability. On the contrary, MNP_OA_DOX is stable at pH 3. The decrease in the size may be explained by the DOX release. No precipitate of MNP occurred after 48 h in the MNP_OA_DOX sample.

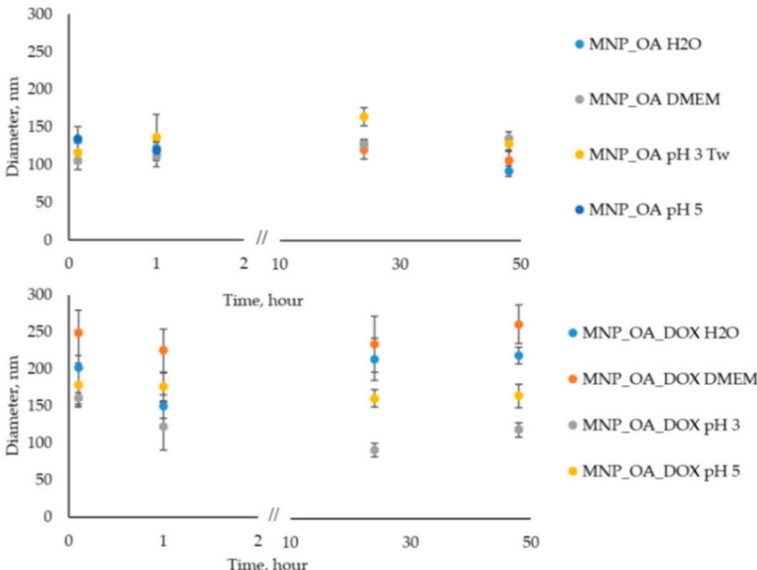

**Figure 6.** Hydrodynamic size variation of 0.13 mg/mL MNP_OA and MNP_OA_DOX evaluated by DLS for 48 h in 100 mM acetic buffer (pH 3 and 5), deionized water (Milli-Q), and cell medium DMEM (100%). For the MNP_OA sample at pH 3 Tw20 (0.5% w), surfactant was added to improve stability.

## 2.5. Anticancer Drug Doxorubicin Release

The DOX release efficiency from the nanomaterials is one of the important factors that can show the drug-loaded MNPs' potential. The release of DOX from MNPs was conducted in a series of the buffers with different pHs (Figure 7). The concentration of the released DOX was measured spectrophotometrically and by fluorescence using a Clariostar plate reader (BMG Labtech, Ortenberg, Germany) (see Section 3.9). The pH range was chosen from 7.4 (mimicking the physiological plasma pH) to acidic, which can be found in the cancer tissue microenvironment and cell endosomes (pH ~ 5). Figure 7 shows the DOX release efficiency from MNPs.

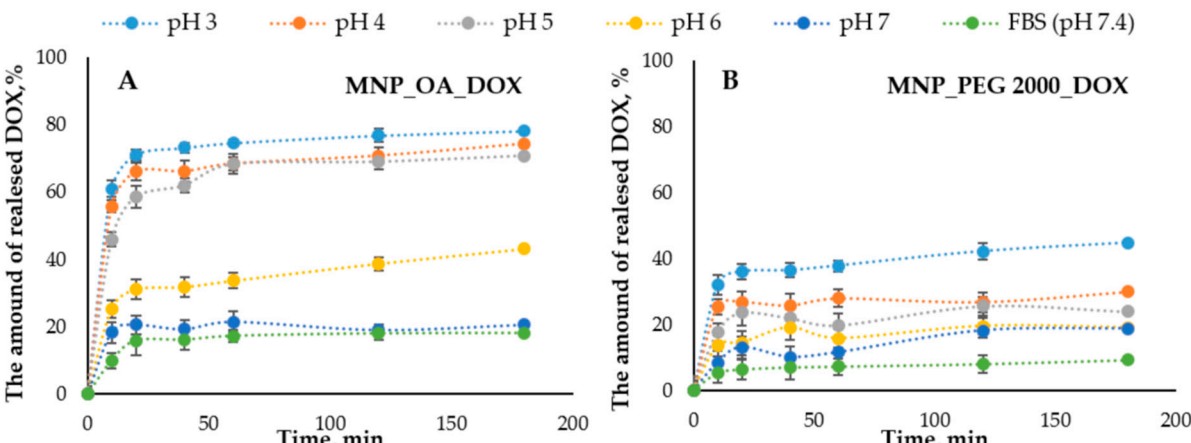

**Figure 7.** The DOX release percent versus time graphs in the pH 3.0–7.4 range. (**A**) MNP_OA_DOX and (**B**) MNP_PEG 2000_DOX.

The nanoparticles show the pH-dependent drug distribution. The general trend is that in alkali conditions, the release is low. In acidic pH, it is much higher. This may be due to the higher stability and solubility of DOX in an acidic medium compared to an alkaline medium [76–78]. Moreover, the protonation of DOX at acidic pH increases its solubility in an aqueous medium, facilitating its release. MNP_OA_DOX shows a low DOX release at pH 7.4, which is much higher under acidic pH (Figure 7). For the efficient

pH-responsible drug release in the tumor tissue, the drug release at about pH 5 has to occur. The pH 3 shows only the extremal results, which are not implemented in the cells. Under such conditions, some additional effects such as magnetic core destabilization or carrier collapse may occur. However, MNP_PEG 2000 has low potential in the pH range (from 3 to 7.4). The efficiency of DOX release from the MNP_OA_DOX is two times higher than for the MNP_PEG 2000_DOX. For this reason, MNP_OA_DOX was chosen for further cell viability studies.

### 2.6. Toxicity Study of MNP_OA_DOX

The degradation of MNPs in the organism may lead to an imbalance in iron metabolism and reactive oxygen species formation [79]. The toxicity experiment on the cell cultures is the next significant step in drug-loaded MNPs efficiency estimation. The easy cell toxicity assay is a widely used MTT test [80,81]. The cell viability experiment was conducted using the A549 human lung carcinoma cell line. Lung cancer is a leading cause of malignancy-related death worldwide. It accounts for nearly 25% of cancer deaths in 2021. For the MTT assay, different amounts of MNPs were incubated with the cells for 48 h (Figure 8). The typical MNPs' dose for the in vitro or in vivo experiments was used. The MNP_OA is non-toxic in the dosing interval. In Figure 8, MNP_OA revealed a high cell viability (~100% and ~93–95%) after 48 h of incubation. The sample concentration ranges from 0.06 µg/mL to 6.26 µg/mL. The MTT test results and MNP_OA stability experiment in the cell medium experiments (Figure 6) confirm the negligible toxicity and excellent biocompatibility of the MNP_OA.

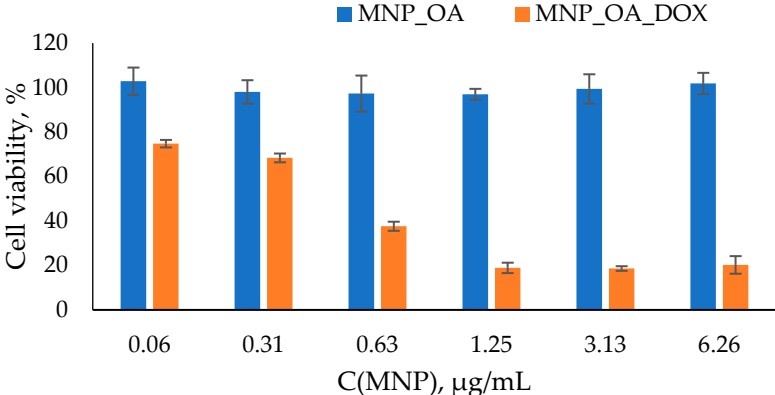

**Figure 8.** Cell viability assay using MTT compound. A549 cell line was incubated for 48 h with MNPs loaded at various concentrations of MNP_OA and MNP_OA_DOX. Cells treated with PBS buffer were used as a 100% viability control.

On the contrary, the drug-loaded MNP_OA_DOX shows high cell suppression efficiency. MNP_OA_DOX with 868 µg/mg (DOX/MNP) capacity was used for the initial cell experiments. The cell viability decreased in a dose-dependent manner to ~20% at the highest concentration (6.26 µg/mL) after 48 h. As displayed in Figures 6 and 7, MNP_OA_DOX stabilizes in the cell medium and has a low drug release at a cell medium with a pH ~ 7. The antitumor activity of MNP_OA_DOX is very high, which cannot be explained by the release of the free DOX into the cell medium. Only the external stimuli DOX release from the MNPs, triggered by acidic pH in the cell endosomes. A low drug concentration of DOX for MNP_OA_DOX and a free DOX have the same toxic effect (Figure S10). With the concentration increase, the DOX toxicity is higher, which is associated with a possible MNPs' prolonged effect and pronounced cell suppression efficiency. After cell penetration, the DOX-loaded MNPs will release the DOX in a time- and pH-dependent manner, which leads to lower acute toxicity. The calculated IC50 value for MNP_OA_DOX is $1.13 \pm 0.02$ mM in terms of DOX concentration and $0.4 \pm 0.03$ µg/mL in terms of the amount of the nanoparticle. It is much lower than the recommended iron oxide nanoparticles' dose (5 mg/kg) for in vivo treatment [16]. The obtained results demonstrate the

possible therapeutic potential of MNP_OA_DOX. However, detailed studies of toxicity, metabolic fate, and immunologic consequences are necessary to establish such drug-loaded nanoparticles for cancer treatment [19,82,83].

## 3. Materials and Methods

### 3.1. Materials

The $FeCl_2 \cdot 4H_2O$ was obtained from Acros organics (99%). The $FeCl_3 \cdot 6H_2O$ was purchased from PanReac AppliChem (97–102%). Polyethylene glycol 2000 (PEG 2000) was acquired from Carl Roth (Karlsruhe, Germany). MTT (3-[4,5-dimethylthiazol-2-yl]-2,5-diphenyl-tetrazolium bromide) was obtained from Panreac Química (Barcelona, Spain), whereas the doxorubicin is from Teva Pharmaceutical Industries Ltd. (Petach Tikva, Israel). Oleic acid, Tween 20, Tween 80, and all solvents and reagents were purchased from Sigma (St. Louis, MO, USA) at the highest available grade and used without purification. In the present work, the following solutions were prepared: sodium borate buffer (SBB, pH 8.5, 0.05 M $Na_2B_4O_7 \cdot 10H_2O$, 0.1 M HCl) and sodium acetate buffer (pH 3–7, 0.2 M NaOAc, 0.2 M AcOH). Deionized water (Milli-Q) was used for the synthesis procedures.

### 3.2. MNPs' Synthesis and Characterization

The one-pot co-precipitation method was used to obtain the iron oxide nanoparticles at room temperature according to a work previously published by our group [70]. Nanoparticles are formed by mixing aqueous solutions of $Fe^{2+}$ and $Fe^{3+}$ salts in a ratio of 1:2 in an acidic solution (HCl). To the mixture, $NH_3$ aqueous solution was added under stirring. The precipitate was washed with perchloric acid. The nanoparticles were stored under nitrogen in deionized water at 4 °C. The nanoparticles were characterized by transmission electron microscopy (TEM) and dynamic light scattering (DLS). The particle size by DLS is 15.6 ± 2.2 nm (PDI = 0.296 ± 0.003). According to TEM data, the size is from 12 to 17 nm. The ζ-potential of MNP is 36.0 ± 0.9 mV (positively charged).

The DLS and ζ-potential measurements were carried out on a Malvern Zetasizer Nano device (Malvern Instruments, Worcestershire, UK) in deionized water (~300 μg/mL concentration). TEM images were conducted on a Jem−1400 device (Jeol, Tokyo, Japan) at an accelerating voltage of 80 kV.

### 3.3. MNP_PEG 2000 Synthesis

The joint synthesis of the PEG 2000-coated MNP was conducted according to the previously described method [84]. Briefly, MNPs' synthesis was carried out by the co-precipitation of iron chloride salts 1.365 g $FeCl_3 \cdot 6$ $H_2O$, 0.5 g $FeCl_2 \cdot 4$ $H_2O$, and 0.15 g PEG 2000 dissolved in 15 mL of 0.17 M HCl. The resulting solution was added dropwise to 15 mL of 1 M NaOH with 0.25 g of PEG 2000 solution preheated to 60 °C. The solution was incubated in a thermoshaker (60 °C, 750 rpm) for 2 h. After the incubation, the particles were separated by magnetic decantation and washed with 30 mL of deionized water three times.

### 3.4. MNP_Tw20 and MNP_Tw80 Synthesis

MNP_Tw20 and MNP_Tw80 synthesis was carried out according to the adapted procedure [70] (par 3.2). The only difference is an addition of Tw20 or Tw80 in the final mass amount of 0.05% in the reaction mixture.

### 3.5. MNP_OA Synthesis

Co-synthesis of magnetic nanoparticles with oleic acid was carried out by the method of co-precipitation of iron chloride salts, $FeCl_2$ and $FeCl_3$, in three successive stages [75]. Briefly, in the first stage, 2.43 g $FeCl_3 \cdot 6$ $H_2O$ and 1.2 g $FeCl_2 \cdot 4$ $H_2O$ were dissolved in 20 mL $H_2O$ and heated to 90 °C. The second stage is the rapid sequential addition of 6 mL of 30% $NH_3$ and 0.4 g of oleic acid to the reaction mixture. The resulting solution was

incubated at 90 °C for 3 h. The precipitate was collected using magnetic decantation and was washed with deionized water until it had neutral pH values.

### 3.6. Reactive Oxygen Species Generation

Reactive oxygen species generation was measured using a methylene blue decolorization assay in a 96-well plate at 25 °C [73]. The samples (100 μL) were prepared by diluting stock concentrations of methylene blue (MB) to 5 μg/mL, hydrogen peroxide to 0.245 M, and MNPs to 37.5, 75, and 150 μg/mL. The concentration of MB was measured by UV-vis spectroscopy at λ 665 nm every 2 min for 80 min using a Clariostar plate reader (BMG Labtech, Ortenberg, Germany). The reaction of MB with hydrogen peroxide was used as a control.

### 3.7. Cytotoxicity Assay (MTT Test)

Tumor cell line from lung adenocarcinoma A549 was cultured in Dulbecco's Minimum Essential Medium (DMEM) supplemented with 10% fetal bovine serum (FBS) (Invitrogen), penicillin (100 units/mL), and streptomycin (100 μg/mL) in a humidified at $37.0 \pm 1.0$ °C, $5.0 \pm 0.5\%$ $CO_2$ incubator in a humid atmosphere. Exponentially growing cells were plated in a 96-well plate ($2 \times 10^3 \pm 0.5 \times 10^3$ cells per well). After overnight incubation, the cells were treated with media containing MNPs (from 0.1 to 10 μg/mL) and incubated for 48 h at a temperature of $37.0 \pm 1.0$ °C in a $CO_2$ atmosphere.

The inhibition of cell proliferation was investigated using a colorimetric assay based on the cleavage of 3-[4,5-dimethylthiazol-2-yl]-2,5-diphenyl-tetrazolium bromide (MTT) by mitochondrial dehydrogenases in viable cells, leading to blue precipitate of formazan formation [80]. To each well, 200 μL of MTT solution (25 mg/mL in RPMI 1640) was added, and the plates were incubated at 37 °C for 3 h. The medium was removed, and formazan was dissolved in 0.1 mL of DMSO. The absorbance at 570 nm (peak) and 620 nm (baseline) was read using a microplate reader Multiscan EX (Thermo Electron Corporation, Waltham, MA, USA). The results were expressed as a percentage of the control values. All values are given as the mean ± standard deviation (SD) values. All measurements were repeated not less than three times.

### 3.8. Doxorubicine-Loaded MNPs

The loading of doxorubicin (DOX, 1 mg/mL) on the MNPs (1 mg) was carried out in 1 mL of sodium borate buffer (SBB, 10 mM, pH 8.5). The mixture was incubated for 12 h at room temperature. Afterward, the MNPs were magnetically precipitated, and the solution was completely withdrawn. The MNPs were washed with one mL of buffer solution SBB buffer twice. The DOX amount in the SBB solution was measured spectrophotometrically ($\lambda = 480$ nm, $\varepsilon = 10{,}410$ $M^{-1}$ $cm^{-1}$). The difference in the DOX concentration in the initial solution and supernatant was used to determine the amount of the drug remaining on the MNPs.

For the MNP_OA capacity of the nanoparticles' dose experiments, 0.5 mL of DOX solution (1 mg/mL) per MNPs was used.

### 3.9. Doxorubicin Release from MNPs

The release of DOX from 1 mg DOX-loaded MNPs was measured in 1 mL of 100 mM acetate buffer solution (pH from 3 to 7) at room temperature under stirring (750 rpm). At various time points (10, 20, 40, 60, 120, and 180 min), the MNPs were magnetically precipitated, and the aliquot (100 μL) of the solution was collected, followed by re-mixing of MNPs. The concentration of the released DOX was measured spectrophotometrically ($\lambda = 480$ nm, $\varepsilon = 10{,}410$ $M^{-1}$ $cm^{-1}$) and by fluorescence ($\lambda_{ex} = 480$ nm, $\lambda_{em} = 590$ nm) using a Clariostar plate reader (BMG Labtech, Ortenberg, Germany). The concentration of the released DOX was calculated using a serial dilution of a DOX standard solution.

## 4. Conclusions

In this study, we presented a novel and smart pH-responsible drug release contraction based on MNPs. We investigated the possible magnetic core stabilization using the co-precipitation MNPS synthesis method and various surfactants. Oleic acid, Tween 20 and 80, and polyethylene glycol 2000 are among them. The synthesized MNPs were characterized by TEM and DLS. The characterization results demonstrate high size agglomerates formed by MNP_Tw nanoparticles. The same MNP_Tw samples have low perspectives due to ROS formation, which was detected by the methylene blue decolorization assay. The remaining MNPs drug loading increased from negligible for non-coated control MNP to 600 µg DOX/1 mg MNP_PEG and up to 870 µg DOX/1 mg for MNP_OA with a high drug loading efficiency of 58% and 87%, respectively. Another proposed method provides a much higher capacity (1757 µg DOX/1 mg MNP_OA) with 24% DOX loading efficiency. In the tested concentration range of 0.06–6.3 µg/mL, MNP_OA demonstrated no toxicity to the cancer cell lines of A549. At the same amount, MNP_OA_DOX demonstrated effective suppression of cancer cell growth. The IC50 values were calculated as $1.13 \pm 0.02$ mM in terms of doxorubicin and $0.4 \pm 0.03$ µg/mL in terms of the nanoparticles' amount. Using the model drug DOX in the MTT assay, we proved that the MNP_OA_DOX agent is a highly potential candidate for constructing a transporter with a prolonged pH-dependent drug-release ability. MNP_OA and MNP_OA_DOX demonstrate high stability in neutral pH, cell medium, and pH 5, which covers the biological aqueous acidity range. MNP_OA has a drug loading ability under high pH values and an efficient drug release at acidic pH (similar to that found in the tumor environment). Due to the small size, magnetic properties, and high drug loading capacity, MNP_OA_DOX are suitable for the future theranostics of cancer.

**Supplementary Materials:** The following supporting information can be downloaded at: https://www.mdpi.com/article/10.3390/magnetochemistry8050054/s1. Figure S1. DLS size distribution data for MNP without coating. The particle size by DLS is $15.6 \pm 2.2$ nm (PDI = $0.296 \pm 0.003$). Figure S2. DLS size distribution (number) of MNP_Tw20. The particle size was $258 \pm 4$ nm (PDI = $0.44 \pm 0.01$). Figure S3. DLS size distribution (intensity) of MNP_Tw20. The particle size was $711 \pm 53$ nm. Figure S4. DLS size distribution (number) of MNP_Tw80. The particle size was $233 \pm 2$ nm (PDI = $0.256 \pm 0.007$). Figure S5. DLS size distribution (intensity) of MNP_Tw80. The intensity particle was $403 \pm 48$ nm. Figure S6. DLS size distribution (number) of MNP_PEG 2000. The particle size was $196 \pm 15$ nm (PDI = $0.50 \pm 0.03$). Figure S7. DLS size distribution (intensity) of MNP_PEG 2000. The intensity particle was $711 \pm 112$ nm. Figure S8. DLS size distribution (number) of MNP_OA. The particle size was $112 \pm 19$ nm (PDI = $0.172 \pm 0.009$). Figure S9. DLS size distribution (intensity) of MNP_OA. The intensity particle was $264 \pm 9$ nm. Figure S10. Cell viability studies of MNP_OA_DOX using MTT assay. A549 cell line was incubated for 48 h with MNPs loaded at various concentrations of MNP_OA_DOX and DOX. Table S1. Hydrodynamic diameter (nm) for MNP_OA for 48 h under various buffer conditions. Table S2. PDI for MNP_OA for 48 h under various buffer conditions. Table S3. Hydrodynamic diameter (nm) for MNP_OA_DOX for 48 h under various buffer conditions. Table S4. PDI for MNP_OA_DOX for 48 h under various buffer conditions. Figure S11. Methylene blue (MB) degradation assay using $H_2O_2$ as an oxidizing agent. The MB preservation % was calculated from the absorbance decrease at 665 nm using a Clariostar plate reader (BMG Labtech, Germany). This graph shows the degradation of MB for all types of nanoparticles at a concentration of 75 µg/mL over the entire period. The percentage of dye binding was subtracted from the percentage of degradation of MB in the reaction with MNP and $H_2O_2$.

**Author Contributions:** Data curation and investigation, E.K., A.C. and E.D.; conceptualization, E.D. and A.C.; writing—original paper, E.K. and A.C.; funding acquisition, A.C. and E.D.; project administration, A.C. All authors have read and agreed to the published version of the manuscript.

**Funding:** This study was funded by the Russian Science Foundation (grant no. 21-74-00120) and the cell experiments partially by the Ministry of Science and Higher Education of the Russian Federation (state registration no. 121031300042-1).

**Institutional Review Board Statement:** Not applicable.

**Informed Consent Statement:** Not applicable.

**Data Availability Statement:** Not available.

**Acknowledgments:** The authors are grateful to Koval O. A. and Nushtaeva A. A. (ICBFM SB RAS, Novosibirsk) for providing the human lung carcinoma cell line. We thank Poletaeva Yu. E. and Ryabchikova E. I. (ICBFM SB RAS) for fruitful discussion and support with TEM images.

**Conflicts of Interest:** The authors declare no conflict of interest.

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
