# Peer review of "High Drug Capacity Doxorubicin-Loaded Iron Oxide Nanocomposites for Cancer Therapy"

_magnetochemistry, doi:10.3390/magnetochemistry8050054_

Round 1

Reviewer 1 Report

The submitted manuscript describes the development of iron oxide nanoparticles coated with three different kinds of surfactants and their load with the anticancer drug doxorubicin in order to be tested as potential agents for anticancer treatment. Although the novelty of the study is not high there are some significant results that should be spread to public to support this scientific direction. I would recommend publication after improvement of the following points.

The motivation of this study should be clearly demonstrated in the introduction. Particularly, introduction should include more information on the state of the art of the specific field while at the end it maust be clarified what is the novel thing added by the current work.

The quality of TEM images is poor. The particle diameters derived by such observation are doubtful. Figure 2E shows nothing at all.

The affinity of the surfactants and especially of oleic acid into water solutions must be explained. To my opinion oleic acid coated nanoparticles cannot be stabilized into water and that makes all experiment related to ROS tests, cytotoxicity assays and drug release in water environment rather questionable.

At low pH values the degradation of iron oxide may occur. Authors should provide information on the stability of nanoparticles when the release of drug at low pH was tested.

Author Response

The authors gratefully thank the Referee for the constructive comments and recommendations, which help to improve the readability and quality of the paper. All the comments are addressed accordingly and have been incorporated into the revised manuscript. Detailed responses to the comments and recommendations are as follows. Please note that all the comments are bold-faced, and the authors' reply follows immediately below the comments. The significant changes in the paper text are highlighted in yellow in the PDF file.

The motivation of this study should be clearly demonstrated in the introduction. Particularly, introduction should include more information on the state of the art of the specific field while at the end it maust be clarified what is the novel thing added by the current work.

We have improved the Introduction section and present much more information about doxorubicin-loaded magnetic nanoparticles. Moreover, we changed the end of the Introduction section and Conclusion section according to the reviewers’ recommendations.

The quality of TEM images is poor. The particle diameters derived by such observation are doubtful. Figure 2E shows nothing at all.

We inserted new TEM images. Groups of the images in Figure 2D, E, and F, Figure 2A, B, and C show the same pictures with different magnification. Low magnification images 2E and 2B need to show the whole material properties but not the MNPs size.

The affinity of the surfactants and especially of oleic acid into water solutions must be explained. To my opinion oleic acid coated nanoparticles cannot be stabilized into water and that makes all experiment related to ROS tests, cytotoxicity assays and drug release in water environment rather questionable.

We have inserted new section 2.5 about MNP_OA and MNP_OA_DOX Stability in Aqueous Solution. The results show the good stability of MNP_OA in an aqueous solution.

At low pH values the degradation of iron oxide may occur. Authors should provide information on the stability of nanoparticles when the release of drug at low pH was tested.

We have inserted new section 2.5 about MNP_OA and MNP_OA_DOX Stability in Aqueous Solution. The results show the good stability of MNP_OA in an aqueous solution. We don’t need pH 3 for biological experiments. Only, pH>5 is relevant for cancer cells and tumor tissue. At this pH, some degradation of the magnetic core may occur. The pH 3 data present in the paper show extremely condition for theoretical investigation of MNPs properties.

Author Response

The authors gratefully thank the Referee for the constructive comments and recommendations, which help to improve the readability and quality of the paper. All the comments are addressed accordingly and have been incorporated into the revised manuscript. Detailed responses to the comments and recommendations are as follows. Please note that all the comments are bold-faced, and the authors' reply follows immediately below the comments. The significant changes in the paper text are highlighted in yellow in the PDF file.

Especially regarding the novelty of the work, I have got concerns. DOX loaded SPIONs with various surfactants, as the authors point out correctly, have been reported many times in the literature. Emphasize the novelty in this work in the manuscript.

We have improved the Introduction section and present much more information about doxorubicin-loaded magnetic nanoparticles. Moreover, we changed the end of the Introduction section and Conclusion section according to the reviewers’ recommendations.

 - line 9 as well as line 34: please change ferromagnetic to ferrimagnetic as magnetite is a ferrimagnetic material. It becomes superparamagnetic if its size is reduced below a certain threshold, but it is not ferromagnetic.

- line 12: please change “three types” to “four types” as you are listing four surfactants later in the sentence. Tw20 and Tw80 are similar, but still different molecules.

- Figure 1: Please add the addition of the base in your reaction scheme. Is it added before or after the surfactants are present.

- Caption of Figure 2: Please do not use “monodisperse”. In the images one can clearly see the different sizes of the MNP cores. Further DLS and PDI measurements do not show monodisperse particles (PDI of 0.5).

- Figure 3: Change mkg/ml in µg/ml.

- Line 102: What are electrically dense nanoparticles? The TEM image shows a high electron density where the nanoparticles are located. This is probably what you mean. Please rephrase that sentence.

- Line 333: Please do not use monodispersity.

We would like to express our gratitude to reviewer’ helpful comment to improve our paper. We have done all this text corrections.

- Figure 2: Please redo the scalebars manually. A, B, D, E it is way too small to read. In C and F it is distorted and wrong. Especially in F: if the scale bar is 100 nm, the particles cannot be 101 nm because the measured distance is much more than the scalebar shows.

We inserted new TEM images. Groups of the images in Figure 2D, E, and F, Figure 2A, B, and C show the same pictures with different magnification.

- Figure S1-S5: Please show the intensity weighted curves of the DLS measurements as they will probably correlate better to the size and PDI values given in Table 1.

We inserted the Intensity size distribution in SI according to the reviewer’s recommendations.

- Line 112: Please define “good colloidal stability”. Stability over one day? One week? One month? Add data or images to show the stability.

We have inserted new section 2.5 about MNP_OA and MNP_OA_DOX Stability in Aqueous Solution. The results show the good stability of MNP_OA in an aqueous solution for not less than 3 weeks.

- Line 170: Is there physical characterization of the particles after the drug loading? How large are the particles? How stable are they? Stability in cell culture media? Please provide data on these questions.

- Figure 6: Did you check the colloidal stability of the different SPIONs in the cell culture medium. If the SPIONs are not stable, they will precipitate onto the cells and "crush" them. Therefore, the cells might turn necrotic. This might only be the chase for DOX loaded SPIONs if they are less stable than MNP_OA in the medium. Please provide stability data for the particles in the cell culture medium.

We have inserted new section 2.5 about MNP_OA and MNP_OA_DOX Stability in Aqueous Solution. MNP_OA_DOX characteristics are presented in Table 3. MNP_OA_DOX is stable in the cell medium.

- Line 202: The MTT assay is an easy test, yes, but nanomaterials can interfere with it and therefore have an influence on the viability data gathered [1,2]. Thus, viability in combination with MNPs should mainly be analyzed by e.g. FACS measurements. Furthermore, DOX can interfere with the assay as well. It still absorbs light at around 570 nm.

The MTT test is the first cell toxicity experiment. Of course, extended toxicity research is required. Flow Cytometry (FACS) Fluorescence Measurement usually requires the fluorescence signal to sort the cells with the conjugate. That is why we need to modify the construction with fluorescence dye. Moreover, apoptosis kits are required for the analysis. However, in the present paper, we consider that the MTT test and ROS quantification are enough to show the absence of severe toxicity and cell inhibition by MNP_OA_DOX.

For the MTT test, to each well, the MTT solution adds, and the plate incubates for several hours. But after it, the formazan precipitate forms. The cells and medium are removed. That is why the DOX amount in the well will be negligible. Moreover, the initial amount of the DOX is extremely low in comparison with MTT. The formazan product has a much higher extinction coefficient and amount in the present conditions.

- Line 190: “That is why under high pH interaction with MNPs surface is preferable.” This means that DOX is not loaded into the surfactant layer but onto the iron oxide surface? If this is the case, the reason for the DOX release at lower pH values could be an acidic degeneration of the particles?

We have changed the discussion in Sections 2.3.      Anticancer Drug Doxorubicin Loading and 2.5 Anticancer Drug Doxorubicin Release. We have inserted some speculation of the mechanism. At the moment, the clear mechanism of DOX loading is not known. The extended research is required. However, the nanoparticles are stable at pH ~ 5 (Figure 6). At pH ~ 3 it can be possible.

- Line 212: As you mentioned before, the release of DOX is highly reduced in neutral and alkaline solutions. As the cell medium is probably at neutral pH, one should not expect a strong release of DOX. That might be the reason why DOX on its own is more efficient. After 48 h single cells should experience a high influence of the loaded particles, which they do not do. And why is the viability increasing with more DOX at a certain point? Can you explain this phenomenon?

The extended DOX release will be not in the cell medium but the endosomal compartment of the cell (pH ~ 5). Some time is required for the MNP_OA_DOX cell penetration and drug release. Figure 8 shows the effect of the MNP_OA_DOX in a dose-dependent manner. The increase of the MNP_OA_DOX leads to a decrease in cell viability. However, it is extremely difficult to suppress the whole-cell pull in the nutrient medium. The amount of the cells grows, and the amount of the MNP_OA_DOX is consumed. That is why no ultimate cell inhibition occurs in the MTT test. Еhe viability increasing with more DOX not known, but it is a reproducible effect. It may be because of the DOX's low solubility and possible precipitation.

Reviewer 3 Report

The submitted paper focuses on an interesting an important topic related to smart drug delivery system. However, the study is well-built and a result provide appropriate information, there are some key issue, which should be revised:

  1. The title of the paper is too general and does not cover the main aim of the research,
  2. the introduction is not detailed enough, it is very rough. There several published researched related to doxorubicin loaded MNPs. Author should mention these papers in details and make a comparison between them and the here reported study. The novelty of this work is missing from here,

  3. the preparation of MNPs via co-precipitation is well-known, why these methods had been selected for the particle production. Authors should highlight the pros and cons of this approach,

  4. the quality of TEM images is not good enough, it should be replaced,

  5. in Table 2, first line: How the DOX/MNP 1 ± 1 content ug/mg could be interpreted? Does it mean that the DOX content was not detectable? The term “1 ± 1” is confusing,

  6. in Fig5 values at zero point (0 min) should be also added to the curves. Panel A and B is missing from the images.

Author Response

The authors gratefully thank the Referee for the constructive comments and recommendations, which help to improve the readability and quality of the paper. All the comments are addressed accordingly and have been incorporated into the revised manuscript. Detailed responses to the comments and recommendations are as follows. Please note that all the comments are bold-faced, and the authors' reply follows immediately below the comments. The significant changes in the paper text are highlighted in yellow in the PDF file.

 The title of the paper is too general and does not cover the main aim of the research,

  1. the introduction is not detailed enough, it is very rough. There several published researched related to doxorubicin loaded MNPs. Author should mention these papers in details and make a comparison between them and the here reported study. The novelty of this work is missing from here,

We have improved the paper title, introduction section and present there much more information about doxorubicin-loaded magnetic nanoparticles. Moreover, we changed the end of the Introduction section and Conclusion section according to the reviewers’ recommendations.

 the preparation of MNPs via co-precipitation is well-known, why these methods had been selected for the particle production. Authors should highlight the pros and cons of this approach,

We presented all synthetic procedures in the Experimental part (par. 3.3.-3.5). Also, we include some information and limitations of co-precipitation method in section 2.1 and 2.4

 The quality of TEM images is not good enough, it should be replaced,

We inserted new TEM images. Groups of the images in Figure 2D, E, and F, Figure 2A, B, and C show the same pictures with different magnification. Low magnification images 2E and 2B need to show the whole material properties but not the MNPs size.

in Table 2, first line: How the DOX/MNP 1 ± 1 content ug/mg could be interpreted? Does it mean that the DOX content was not detectable? The term “1 ± 1” is confusing,

We have changed this value to n.d. – not determined, negligible amount of DOX.
in Fig5 values at zero point (0 min) should be also added to the curves. Panel A and B is missing from the images.

 We have changed the picture according to referee’s recommendation.

Round 2

Reviewer 1 Report

The response from authors was reasonable and covers satisfactorily the comments. I would recommend publication.

Reviewer 2 Report

Dear authors,

the current version of the manuscript with the included corrections is suitable for publication in the journal.

One small correction still needs to be done: Please change ferromagnetic to ferrimagnetic in line 10 as well as Fe3O4 to Fe3O4.

Reviewer 3 Report

Authors gave satisfied answers

and did the suggested corrections and editing issues.